

# Complex characterization of oat (*Avena sativa* L.) lines obtained by wide crossing with maize (*Zea mays* L.)

Edyta Skrzypek[1], Tomasz Warzecha[2], Angelika Noga[1], Marzena Warchoł[1], Ilona Czyczyło-Mysza[1], Kinga Dziurka[1], Izabela Marcińska[1], Kamila Kapłoniak[1], Agnieszka Sutkowska[2], Zygmunt Nita[3], Krystyna Werwińska[3], Dominika Idziak-Helmcke[4], Magdalena Rojek[4] and Marta Hosiawa-Barańska[4]

[1] Department of Biotechnology, Polish Academy of Sciences, The Franciszek Górski Institute of Plant Physiology, Kraków, Poland
[2] Department of Plant Breeding and Seed Science, University of Agriculture, Kraków, Polska
[3] Plant Breeding Strzelce Ltd., PBAI Group, Strzelce, Polska
[4] Department of Plant Anatomy and Cytology, University of Silesia in Katowice, Katowice, Polska

Corresponding author
Edyta Skrzypek, e.skrzypek@ifr-pan.edu.pl

## ABSTRACT

**Background**. The oat × maize addition (OMA) lines are used for mapping of the maize genome, the studies of centromere-specific histone (CENH3), gene expression, meiotic chromosome behavior and also for introducing maize C4 photosynthetic system to oat. The aim of our study was the identification and molecular-cytogenetic characterization of oat × maize hybrids.

**Methods**. Oat DH lines and oat × maize hybrids were obtained using the wide crossing of *Avena sativa* L. with *Zea mays* L. The plants identified as having a *Grande-1* retrotransposon fragment, which produced seeds, were used for genomic *in situ* hybridization (GISH).

**Results**. A total of 138 oat lines obtained by crossing of 2,314 oat plants from 80 genotypes with maize cv. Waza were tested for the presence of maize chromosomes. The presence of maize chromatin was indicated in 66 lines by amplification of the PCR product (500 bp) generated using primers specific for the maize retrotransposon *Grande-1*. Genomic *in situ* hybridization (GISH) detected whole maize chromosomes in eight lines (40%). All of the analyzed plants possessed full complement of oat chromosomes. The number of maize chromosomes differed between the OMA lines. Four OMA lines possessed two maize chromosomes similar in size, three OMA—one maize chromosome, and one OMA—four maize chromosomes. In most of the lines, the detected chromosomes were labeled uniformly. The presence of six 45S rDNA loci was detected in oat chromosomes, but none of the added maize chromosomes in any of the lines carried 45S rDNA locus. Twenty of the analyzed lines did not possess whole maize chromosomes, but the introgression of maize chromatin in the oat chromosomes. Five of 66 hybrids were shorter in height, grassy type without panicles. Twenty-seven OMA lines were fertile and produced seeds ranging in number from 1–102 (in total 613). Sixty-three fertile DH lines, out of 72 which did not have an addition of maize chromosomes or chromatin, produced seeds in the range of 1–343 (in total 3,758). Obtained DH and OMA lines were fertile and produced seeds.

**Discussion**. In wide hybridization of oat with maize, the complete or incomplete chromosomes elimination of maize occur. Hybrids of oat and maize had a complete

set of oat chromosomes without maize chromosomes, and a complete set of oat chromosomes with one to four retained maize chromosomes.

# INTRODUCTION

The fusion of genomes from various genetic backgrounds through wide hybridization is essential for plant breeders seeking new crop cultivars, especially in the context of climate changes. On the other hand, unlikely hybrids with genomes from both the parental species are not always obtained. Uniparental chromosome elimination, the removal of the one parent chromosomes from the wide hybrids, might be a benefit to the breeders for fast genetic improvement of the crop cultivars (*Chaudhary et al., 2013*).

In crosses between Panicoideae and Pooideae, in hybrids e.g., wheat × sorghum, wheat × pearl millet, wheat × maize, oat × maize or oat × pearl millet, either Panicoideae chromosomes are eliminated shortly after fertilization at the beginning of embryogenesis, or incomplete chromosomes elimination occurred (*Laurie & Bennett, 1986*; *Laurie & Bennett, 1988*; *Laurie, 1989*; *Rines & Dahleen, 1990*). In the case of maize crosses, the elimination of maize chromosomes was due to their limited mobility for the duration of mitotic metaphase and anaphase as a result of the failure of spindle fiber attachment to the centromeres (*Laurie & Bennett, 1989*).

Chromosome elimination of the alien genome after fertilization of the different species can take place in various distant hybrids (*Dunwell, 2010*). Interspecific hybridization was the first observed in the genus Hordeum, during crossing cultivated barley (*Hordeum vulgare* L.) as the female with wild species *Hordeum bulbosum* L. (*Kasha & Kao, 1970*). In an early stage of development of a hybrid embryo, chromosomes of the female are eliminated and developed haploid embryos contain only chromosomes of cultivated barley (*Liu et al., 2014*). Chromosome elimination has been described in crosses of *Triticum aestivum* and Triticeae species (*H. vulgare* and *H. bulbosum*) as well as more distantly related species (*Zea mays*, *Pennisetum glaucum*, *Sorghum bicolor*, *Coix lacryma-jobi*, *Imperata cylindrica*) (*Barclay, 1975*; *Laurie & Bennett, 1986*; *Laurie & Bennett, 1988*; *Laurie, 1989*; *Inagaki & Mujeeb-Kazi, 1995*; *Mochida & Tsujimoto, 2001*; *Gernand et al., 2005*; *Komeda et al., 2007*; *Chaudhary et al., 2013*).

A new situation was observed in the crosses between oat (*Avena sativa* L.) and maize (*Zea mays* L.) where oat × maize hybrid embryos developed into euhaploid plants with the whole oat chromosomes without maize chromosomes and into aneuhaploid plants with the complete set of oat chromosome and added a different numbers of maize chromosomes (*Kynast et al., 2012*). In the aneuhaploid plants numerous maize chromosomes were not removed during embryogenesis, and the preserved maize chromosomes ultimately were stable and act as oat chromosomes in mitosis. The consequence of it was obtaining oat × maize addition (OMA) lines.

The first sweet maize line used to obtain OMA was the Seneca 60. In oat was identified a fragment of Seneca 60 chromosome 3 containing a neocentromere (*Topp et al., 2009*). Over the past few years, fertile disomic OMA of diverse oat cultivars have been reported, in which *Zea mays* was the recipient of chromosomes 1–7 and 9, the short arm of 10, as well as a monosomic OMA with chromosome 8 (*Rines et al., 2009*). Amongst maize crosses with other cereals, only oat × maize hybrids formed embryos which are able to germinate and grow into partially fertile plants. In these OMA lines the partial fertility is a result of meiotic restitution, which improved the production of unreduced gametes. The observation of haploid oat microsporocytes (with or without maize chromosomes) displayed that in microsporogenesis karyokinesis is irregular, mainly in meiosis (*Kynast et al., 2012*).

*Ishii et al. (2013)* reported that in crosses of oat × pearl millet (*Pennisetum glaucum* L.), chromosome and marker analyses showed that the seedlings were hybrids that had all oat and some pearl millet chromosomes. Twenty-one embryos in this study germinated and showed only shoot growth which turned to the scutellum and finally died under light regime.

The OMA lines are irreplaceable and valuable genetic materials, where the heterologous organism is required to investigate the role and structure of single maize chromosome, to ascribe and localize molecular markers and genes on maize chromosomes, or to study the dynamic of chromosome pairing. Dissecting the maize chromosomes by OMA lines can deliver plants with maize subchromosomal fragments. As OMA lines, subchromosomal materials are used in a series of genomics and chromosome studies (*Kynast & Riera-Lizarazu, 2011*).

The OMA lines were used in the cytogenetic mapping system for maize genome (*Riera-Lizarazu, Rines & Phillips, 1996*; *Maquieira, 1997*; *Kynast et al., 2001*; *Kynast et al., 2002*; *Okagaki et al., 2001*; *Koumbaris & Bass, 2003*), gene expression (*Jin et al., 2004*), meiotic chromosome arrangements (*Bass et al., 2000*), investigation of conservation of gene expression of prolamin storage protein (*Garcia et al., 2015*) as well as investigation of genes transcriptional regulation in new genomic environments (*Dong et al., 2018*). It was shown that the majority of maize genes displayed maize-specific transcription in the oat genomic environment. Oat × maize hybrids were also used for identification (*Topp et al., 2009*) and analyzing the size and location of maize chromosomes centromeres transferred into oat by immunoprecipitation for the histone CENH3 (*Wang et al., 2014*). The mutation of CENH3 offers promising opportunities for application in a wide range of crop species for haploid production (*Ravi & Chan, 2010*; *Ishii et al., 2015*; *Karimi-Ashtiyani et al., 2015*; *Maheshwari et al., 2015*; *Britt & Kuppu, 2016*; *Kelliher et al., 2016*). A single-point amino acid exchange in the centromere-targeting domain of CENH3 leads to reduced centromere loading of CENH3 in barley, sugar beet, and *Arabidopsis thaliana*. Haploids were obtained when *cenh3*-null mutants were crossed with desired plants. The OMA lines were also used in the examination for expression of the $C_4$ photosynthetic system (*Kynast et al., 2001*; *Kowles et al., 2008*; *Dong et al., 2018*).

The aim of the study was the identification and molecular-cytogenetic characterization of oat × maize hybrids. The paper also presents vigor of DH and OMA lines and their effectiveness in seeds production.

## MATERIAL AND METHODS

### Oat wide crossing with maize

The experiments were done on 80 oat genotypes (listed in Table 1) obtained from Plant Breeding Strzelce Ltd., PBAI Group, Strzelce, Łódź Voivodeship, Poland. Oat plants and maize cultivar Waza were sown and cultivated according to *Noga et al. (2016)*. Oat haploid plants were obtained using the wide crossing method by pollination with maize cv. Waza, as described by *Marcińska et al. (2013)*. Three weeks after floret pollination, enlarged ovaries were sterilized in 70% (*v/v*) ethanol (1 min), in a 2.5% (*w/v*) calcium hypochlorite (8 min), and washed three times with sterile water. Then, haploid embryos were isolated from caryopses and transferred to the 190-2 medium (*Zhuang & Xu, 1983*) with 9% (*w/v*) maltose, 0.6% (*w/v*) agar, 0.5 mg L$^{-1}$ kinetin (KIN) and 0.5 mg L$^{-1}$ $\alpha$-naphthaleneacetic acid (NAA) and pH 6.0. Haploid embryos were germinated in an *in vitro* chamber with a 16 h photoperiod, light intensity of 100 $\mu$mol m$^2$ s$^{-1}$ and temperature 21 °C. The germination capacity of haploid embryos was evaluated under a stereomicroscope (SMZ 1500, Nikon, Tokyo, Japan) and photographs were taken using a digital CCD camera (DS-Ri1, Nikon, Tokyo, Japan).

The plants developed from haploid embryos were transferred to MS medium (*Murashige & Skoog, 1962*) with 0.6% (*w/v*) agar. Then, the well rooted plants were acclimated to natural conditions by transferring them subsequently to wet perlite and soil. For chromosome doubling, the roots of haploid plants were treated by colchicine solution and the ploidy level of the plants was evaluated using a MACSQuant flow cytometer (MACSQuant, Miltenyi Biotec GmbH, Bergisch Gladbach, Germany) as described by *Noga et al. (2016)*.

Next, the plants were tested for the presence of the maize retrotransposon *Grande-1* to indicate maize chromatin in oat. Oat cv. Stoper and maize cv. Waza were used as a negative and positive control of the presence of the maize retrotransposon *Grande-1*. Leaves of studied lines, oat cv. Stoper, and maize cv. Waza (about 200 mg) were lyophilized in high vacuum at 40 $\mu$bar, coil temperature $-52$ °C (lyophiliser FreeZone 6L; Labconco, Kansas City, MO, USA). Then, the tissue was homogenized using ball mill (MM400; Retsch, Haan, Germany) for 5 min at frequency 25 Hz.

### DNA extraction and PCR analyses

DNA was extracted from 20 mg of lyophilized leaves, using the Genomic Mini AX Plant kit (A&A Biotechnology, Gdynia, Poland) according to the producer recommendations. After precipitation, the DNA pellet was washed twice in 70% (v/v) ethanol and dried in a concentrator (Concentrator Plus; Eppendorf, Hamburg, Germany) for 10 min. under 20 hPa. The precipitate was suspended in sterile distilled water. DNA concentration was measured at 260 nm with a spectrophotometer NanoDrop 2000c (Thermo Scientific, USA).

Two Grande 1F (5′-AAA GAC CTC ACG AAA GGC CCA AGG-3′), Grande1R (5′-AAA TGG TTC ATG CCG ATT GCA CG-3′) primers (GenBank accession number X97604;

**Table 1** The influence of oat (*Avena sativa L.*) genotype on the efficiency of haploid embryos, plants and seeds production using wide crossing with maize (*Zea mays L.*).

| No | Genotype | No. of plants | No. of emasculated florets | No. of isolated embryos | No. of germinated embryos | No. of acclimated plants | No. of lines | Embryos/ florets (%) | Germinated embryos/ florets (%) | Acclimated plants/ florets (%) | Lines/ florets (%) | No. of seeds | No. of maize chromosome/ chromatin added to oat genome |
|---|---|---|---|---|---|---|---|---|---|---|---|---|---|
| 1 | STH 4.403/1 | 27 | 858 | 11 | 0 | 0 | 0 | 1.28 | 0 | 0 | 0 | 0 [d] | 0 |
| 2 | STH 4.403/2 | 29 | 856 | 23 | 3 | 1 | 1 | 2.69 | 0.35 | 0.12 | 0.12 | 105 [bc] | 0 |
| 3 | STH 4.403/3 | 32 | 984 | 10 | 3 | 1 | 1 | 1.02 | 0.30 | 0.10 | 0.10 | 35 [d] | 2 |
| 4 | STH 4.403/4 | 23 | 763 | 12 | 3 | 2 | 2 | 1.57 | 0.39 | 0.26 | 0.26 | | |
| | | | | | | | STH 4.403/4 a | | | | | 128 [bc] | 0 |
| | | | | | | | STH 4.403/4 b | | | | | 9 [d] | 0 |
| 5 | STH 4.4576 | 26 | 439 | 21 | 9 | 2 | 1 | 4.78 | 2.05 | 0.46 | 0.23 | 10 [d] | chromatin |
| 6 | STH 4.4577 | 33 | 559 | 23 | 6 | 1 | 0 | 4.11 | 1.07 | 0.18 | 0 | 0 [d] | 0 |
| 7 | STH 4.4586 | 40 | 674 | 45 | 21 | 3 | 1 | 6.68 | 3.12 | 0.45 | 0.15 | 0 [d] | n/a* |
| 8 | STH 4.4595 | 40 | 523 | 21 | 13 | 5 | 0 | 4.03 | 2.49 | 0.96 | 0 | 0 [d] | 0 |
| 9 | STH 4.4606 | 39 | 832 | 60 | 34 | 8 | 2 | 7.22 | 4.09 | 0.96 | 0.24 | | |
| | | | | | | | STH 4.4606 a | | | | | 36 [cd] | chromatin |
| | | | | | | | STH 4.4606 b | | | | | 0 [d] | n/a |
| 10 | STH 4.4690 | 45 | 1,094 | 106 | 62 | 20 | 16 | 9.69 | 5.67 | 1.83 | 1.46 | | |
| | | | | | | | STH 4.4690 a | | | | | 19 [d] | 0 |
| | | | | | | | STH 4.4690 b | | | | | 119 [bc] | 0 |
| | | | | | | | STH 4.4690 c | | | | | 23 [d] | 0 |
| | | | | | | | STH 4.4690 d | | | | | 10 [d] | chromatin |
| | | | | | | | STH 4.4690 e | | | | | 12 [d] | 0 |
| | | | | | | | STH 4.4690 f | | | | | 2 [d] | chromatin |
| | | | | | | | STH 4.4690 g | | | | | 200 [bc] | 0 |
| | | | | | | | STH 4.4690 h | | | | | 2 [d] | 0 |
| | | | | | | | STH 4.4690 i | | | | | 6 [d] | 0 |
| | | | | | | | STH 4.4690 j | | | | | 1 [d] | 0 |
| | | | | | | | STH 4.4690 k | | | | | 1 [d] | 0 |
| | | | | | | | STH 4.4690 l | | | | | 0 [d] | 0 |
| | | | | | | | STH 4.4690 m | | | | | 0 [d] | n/a |
| | | | | | | | STH 4.4690 n | | | | | 0 [d] | n/a |
| | | | | | | | STH 4.4690 o | | | | | 0 [d] | n/a |
| | | | | | | | STH 4.4690 p | | | | | 1 [d] | 4 |
| 11 | STH 4.4729 | 36 | 927 | 33 | 4 | 2 | 1 | 3.56 | 0.43 | 0.22 | 0.11 | 0 [d] | n/a |

**Table 1** (*continued*)

| No | Genotype | No. of plants | No. of emasculated florets | No. of isolated embryos | No. of germinated embryos | No. of acclimated plants | No. of lines | Embryos/ florets (%) | Germinated embryos/ florets (%) | Acclimated plants/ florets (%) | Lines/ florets (%) | No. of seeds | No. of maize chromosome/ chromatin added to oat genome |
|----|----------|------|------|------|------|------|------|------|------|------|------|------|------|
| 12 | STH 4.4731 | 35 | 776 | 33 | 13 | 5 | 4 | 4.25 | 1.68 | 0.64 | 0.52 | | |
| | | | | | | | STH 4.4731 a | | | | | 91 [bc] | 0 |
| | | | | | | | STH 4.4731 b | | | | | 14 [d] | 0 |
| | | | | | | | STH 4.4731 c | | | | | 14 [d] | 0 |
| | | | | | | | STH 4.4731 d | | | | | 6 [d] | 0 |
| 13 | STH 4.4740 | 33 | 717 | 38 | 10 | 3 | 2 | 5.30 | 1.39 | 0.42 | 0.28 | | |
| | | | | | | | STH 4.4740 a | | | | | 20 [d] | 0 |
| | | | | | | | STH 4.4740 b | | | | | 57 [bcd] | 0 |
| 14 | STH 4.4742 | 22 | 647 | 34 | 17 | 7 | 4 | 5.26 | 2.63 | 1.08 | 0.62 | | |
| | | | | | | | STH 4.4742 a | | | | | 77 [bc] | chromatin |
| | | | | | | | STH 4.4742 b | | | | | 222 [b] | 0 |
| | | | | | | | STH 4.4742 c | | | | | 5 [d] | 0 |
| | | | | | | | STH 4.4742 d | | | | | 1 [d] | chromatin |
| 15 | STH 4.8402 | 36 | 626 | 31 | 8 | 1 | 0 | 4.95 | 1.28 | 0.16 | 0 | 0 [d] | 0 |
| 16 | STH 4.8411 | 28 | 623 | 27 | 9 | 1 | 1 | 4.33 | 1.44 | 0.16 | 0.16 | 165 [bc] | |
| 17 | STH 4.8417 | 26 | 648 | 28 | 10 | 4 | 4 | 4.32 | 1.54 | 0.62 | 0.62 | | |
| | | | | | | | STH 4.8417 a | | | | | 0 [d] | n/a |
| | | | | | | | STH 4.8417 b | | | | | 0 [d] | n/a |
| | | | | | | | STH 4.8417 c | | | | | 0 [d] | n/a |
| | | | | | | | STH 4.8417 d | | | | | 10 [d] | 0 |
| 18 | STH 4.8432 | 34 | 681 | 26 | 4 | 1 | 1 | 3.82 | 0.59 | 0.15 | 0.15 | 0 [d] | 0 |
| 19 | STH 4.8435 | 35 | 653 | 29 | 5 | 1 | 0 | 4.44 | 0.77 | 0.15 | 0 | 0 [d] | 0 |
| 20 | STH 4.8437 | 30 | 602 | 35 | 10 | 5 | 2 | 5.81 | 1.66 | 0.83 | 0.33 | | |
| | | | | | | | STH 4.8437 a | | | | | 4 [d] | 0 |
| | | | | | | | STH 4.8437 b | | | | | 0 [d] | n/a |
| 21 | STH 4.8442 | 33 | 826 | 45 | 9 | 3 | 1 | 5.45 | 1.09 | 0.36 | 0.12 | 0 [d] | n/a |
| 22 | STH 4.8448 | 29 | 518 | 33 | 11 | 2 | 1 | 6.37 | 2.12 | 0.39 | 0.19 | 0 [d] | n/a |
| 23 | STH 4.8456/1 | 51 | 1,019 | 29 | 3 | 2 | 2 | 2.85 | 0.29 | 0.20 | 0.20 | | |
| | | | | | | | STH 4.8456/1 a | | | | | 7 [d] | 0 |
| | | | | | | | STH 4.8456/1 b | | | | | 6 [d] | 0 |
| 24 | STH 4.8456/2 | 45 | 1,043 | 27 | 10 | 7 | 7 | 2.59 | 0.96 | 0.67 | 0.67 | | |
| | | | | | | | STH 4.8456/2 a | | | | | 80 [bc] | 0 |
| | | | | | | | STH 4.8456/2 b | | | | | 39 [cd] | 0 |
| | | | | | | | STH 4.8456/2 c | | | | | 17 [d] | 0 |

Skrzypek et al. (2018), *PeerJ*, DOI 10.7717/peerj.5107

**Table 1** (*continued*)

| No | Genotype | No. of plants | No. of emasculated florets | No. of isolated embryos | No. of germinated embryos | No. of acclimated plants | No. of lines | Embryos/ florets (%) | Germinated embryos/ florets (%) | Acclimated plants/ florets (%) | Lines/ florets (%) | No. of seeds | No. of maize chromosome/ chromatin added to oat genome |
|----|----------|----|----|----|----|----|----|----|----|----|----|----|----|
| | | | | | | | STH 4.8456/2 d | | | | | 11 [d] | 0 |
| | | | | | | | STH 4.8456/2 e | | | | | 53 [cd] | chromatin |
| | | | | | | | STH 4.8456/2 f | | | | | 1 [d] | chromatin |
| | | | | | | | STH 4.8456/2 g | | | | | 0 [d] | 0 |
| 25 | STH 4.8457/1 | 42 | 887 | 30 | 1 | 1 | 1 | 3.38 | 0.11 | 0.11 | 0.11 | 53 [cd] | 0 |
| 26 | STH 4.8457/2 | 47 | 1,141 | 47 | 8 | 5 | 5 | 4.12 | 0.70 | 0.44 | 0.44 | | |
| | | | | | | | STH 4.8457/2 a | | | | | 105 [bc] | 0 |
| | | | | | | | STH 4.8457/2 b | | | | | 0 [d] | 0 |
| | | | | | | | STH 4.8457/2 c | | | | | 0 [d] | n/a |
| | | | | | | | STH 4.8457/2 d | | | | | 0 [d] | n/a |
| | | | | | | | STH 4.8457/2 e | | | | | 0 [d] | n/a |
| 27 | STH 4.8459 | 20 | 286 | 13 | 2 | 1 | 0 | 4.55 | 0.70 | 0.35 | 0 | 0 [d] | 0 |
| 28 | STH 5.451/1 | 25 | 627 | 10 | 2 | 0 | 0 | 1.59 | 0.32 | 0 | 0 | 0 [d] | 0 |
| 29 | STH 5.451/2 | 23 | 614 | 17 | 5 | 3 | 2 | 2.77 | 0.81 | 0.49 | 0.33 | | |
| | | | | | | | STH 5.451/2 | | | | | 0 [d] | n/a |
| | | | | | | | STH 5.451/2 | | | | | 0 [d] | n/a |
| 30 | STH 5.451/3 | 21 | 509 | 6 | 4 | 1 | 1 | 1.18 | 0.79 | 0.20 | 0.20 | 0 [d] | 0 |
| 31 | STH 5.451/4 | 21 | 518 | 9 | 1 | 0 | 0 | 1.74 | 0.19 | 0 | 0 | 0 [d] | 0 |
| 32 | STH 5.8046 | 2 | 77 | 2 | 0 | 0 | 0 | 2.60 | 0 | 0 | 0 | 0 [d] | 0 |
| 33 | STH 5.8421 | 35 | 998 | 32 | 8 | 7 | 7 | 3.21 | 0.80 | 0.70 | 0.70 | | |
| | | | | | | | STH 5.8421 a | | | | | 183 [bc] | 0 |
| | | | | | | | STH 5.8421 b | | | | | 24 [d] | 0 |
| | | | | | | | STH 5.8421 c | | | | | 20 [d] | 0 |
| | | | | | | | STH 5.8421 d | | | | | 102 [bc] | chromatin |
| | | | | | | | STH 5.8421 e | | | | | 0 [d] | n/a |
| | | | | | | | STH 5.8421 f | | | | | 0 [d] | n/a |
| | | | | | | | STH 5.8421 g | | | | | 0 [d] | n/a |
| 34 | STH 5.8422/1 | 15 | 620 | 7 | 0 | 0 | 0 | 1.13 | 0 | 0 | 0 | 0 [d] | 0 |
| 35 | STH 5.8422/2 | 15 | 487 | 5 | 2 | 0 | 0 | 1.03 | 0 | 0 | 0 | 0 [d] | 0 |
| 36 | STH 5.8423 | 39 | 1,014 | 47 | 8 | 3 | 2 | 4.64 | 0.79 | 0.30 | 0.20 | | |
| | | | | | | | STH 5.8423 a | | | | | 1 [d] | 0 |
| | | | | | | | STH 5.8423 b | | | | | 0 [d] | n/a |
| 37 | STH 5.8424/1 | 22 | 455 | 13 | 5 | 2 | 1 | 2.86 | 1.10 | 0.44 | 0.22 | 21 [d] | 0 |
| 38 | STH 5.8424/2 | 30 | 874 | 44 | 6 | 2 | 1 | 5.03 | 0.69 | 0.23 | 0.11 | 0 [d] | n/a |

Peer]

**Table 1** (*continued*)

| No | Genotype | No. of plants | No. of emasculated florets | No. of isolated embryos | No. of germinated embryos | No. of acclimated plants | No. of lines | Embryos/ florets (%) | Germinated embryos/ florets (%) | Acclimated plants/ florets (%) | Lines/ florets (%) | No. of seeds | No. of maize chromosome/ chromatin added to oat genome |
|---|---|---|---|---|---|---|---|---|---|---|---|---|---|
| 39 | STH 5.8425 | 40 | 1,284 | 84 | 21 | 9 | 5 | 6.54 | 1.64 | 0.70 | 0.39 | | |
| | STH 5.8425 a | | | | | | | | | | | 3 [d] | chromatin |
| | STH 5.8425 b | | | | | | | | | | | 1 [d] | chromatin |
| | STH 5.8425 c | | | | | | | | | | | 0 [d] | n/a |
| | STH 5.8425 d | | | | | | | | | | | 0 [d] | n/a |
| | STH 5.8425 e | | | | | | | | | | | 0 [d] | n/a |
| 40 | STH 5.8426 | 26 | 738 | 33 | 5 | 2 | 1 | 4.47 | 0.68 | 0.27 | 0.14 | 28 [d] | 2 |
| 41 | STH 5.8427 | 31 | 1,145 | 46 | 11 | 6 | 5 | 4.02 | 0.96 | 0.52 | 0.44 | | |
| | STH 5.8427 a | | | | | | | | | | | 211 [bc] | 0 |
| | STH 5.8427 b | | | | | | | | | | | 179 [bc] | 0 |
| | STH 5.8427 c | | | | | | | | | | | 158 [bc] | 0 |
| | STH 5.8427 d | | | | | | | | | | | 65 [bcd] | chromatin |
| | STH 5.8427 e | | | | | | | | | | | 1 [d] | chromatin |
| 42 | STH 5.8428/1 | 3 | 61 | 0 | 0 | 0 | 0 | 0 | 0 | 0 | 0 | 0 [d] | 0 |
| 43 | STH 5.8428/2 | 35 | 690 | 11 | 2 | 1 | 1 | 1.59 | 0 | 0.14 | 0.14 | 4 [d] | chromatin |
| 44 | STH 5.8429 | 34 | 1,253 | 64 | 22 | 7 | 7 | 5.11 | 1.76 | 0.56 | 0.56 | | |
| | STH 5.8429 a | | | | | | | | | | | 343 [a] | 0 |
| | STH 5.8429 b | | | | | | | | | | | 8 [d] | 0 |
| | STH 5.8429 c | | | | | | | | | | | 4 [d] | chromatin |
| | STH 5.8429 d | | | | | | | | | | | 0 [d] | n/a |
| | STH 5.8429 e | | | | | | | | | | | 0 [d] | n/a |
| | STH 5.8429 f | | | | | | | | | | | 0 [d] | n/a |
| | STH 5.8429 g | | | | | | | | | | | 0 [d] | n/a |
| 45 | STH 5.8430/1 | 18 | 351 | 2 | 0 | 0 | 0 | 0.57 | 0 | 0 | 0 | 0 [d] | 0 |
| 46 | STH 5.8430/2 | 17 | 307 | 17 | 3 | 1 | 1 | 5.54 | 0.98 | 0.33 | 0.33 | 130 [bc] | 0 |
| 47 | STH 5.8432 | 32 | 698 | 29 | 4 | 1 | 1 | 4.15 | 0.57 | 0.14 | 0.14 | 58 [cd] | 0 |
| 48 | STH 5.8436 | 39 | 799 | 24 | 3 | 2 | 2 | 3.00 | 0.38 | 0.25 | 0.25 | | |
| | STH 5.8436 a | | | | | | | | | | | 58 [cd] | 0 |
| | STH 5.8436 b | | | | | | | | | | | 8 [d] | 2 |
| 49 | STH 5.8440 | 35 | 817 | 43 | 7 | 2 | 2 | 5.26 | 0.86 | 0.24 | 0.24 | | |
| | STH 5.8440 a | | | | | | | | | | | 96 [bcd] | 0 |
| | STH 5.8440 b | | | | | | | | | | | 51 [cd] | chromatin |
| 50 | STH 5.8449 | 38 | 844 | 21 | 4 | 2 | 2 | 2.49 | 0.47 | 0.24 | 0.24 | | |

**Table 1** (*continued*)

| No | Genotype | No. of plants | No. of emasculated florets | No. of isolated embryos | No. of germinated embryos | No. of acclimated plants | No. of lines | Embryos/florets (%) | Germinated embryos/florets (%) | Acclimated plants/florets (%) | Lines/florets (%) | No. of seeds | No. of maize chromosome/chromatin added to oat genome |
|----|----------|------|------|------|------|------|------|------|------|------|------|------|------|
| | | | | | | | STH 5.8449 a | | | | | 70 [bcd] | 0 |
| | | | | | | | STH 5.8449 b | | | | | 55 [cd] | chromatin |
| 51 | STH 5.8450 | 30 | 628 | 27 | 0 | 0 | 0 | 4.30 | 0 | 0 | 0 | 0 [d] | 0 |
| 52 | STH 5.8458 | 34 | 727 | 27 | 4 | 2 | 2 | 3.71 | 0.55 | 0.28 | 0.28 | | |
| | | | | | | | STH 5.8458 a | | | | | 57 [cd] | 0 |
| | | | | | | | STH 5.8458 b | | | | | 35 [d] | 2 |
| 53 | STH 5.8460 | 25 | 511 | 16 | 1 | 0 | 0 | 3.13 | 0 | 0 | 0 | 0 [d] | 0 |
| 54 | STH 5.8504 | 22 | 840 | 25 | 9 | 4 | 2 | 2.98 | 1.07 | 0.48 | 0.24 | | |
| | | | | | | | STH 5.8504 a | | | | | 2 [d] | 0 |
| | | | | | | | STH 5.8504 b | | | | | 5 [d] | 1 |
| 55 | STH 5.8504/1 | 25 | 808 | 34 | 15 | 7 | 3 | 4.21 | 1.86 | 0.87 | 0.37 | | |
| | | | | | | | STH 5.8504/1 a | | | | | 2 [d] | 0 |
| | | | | | | | STH 5.8504/1 b | | | | | 0 [d] | n/a |
| | | | | | | | STH 5.8504/1 c | | | | | 0 [d] | n/a |
| 56 | STH 5.8505 | 27 | 946 | 33 | 10 | 4 | 3 | 3.49 | 1.06 | 0.42 | 0.32 | | |
| | | | | | | | STH 5.8505 a | | | | | 19 [d] | 0 |
| | | | | | | | STH 5.8505 b | | | | | 0 [d] | n/a |
| | | | | | | | STH 5.8505 c | | | | | 0 [d] | n/a |
| 57 | STH 5.8505/1 | 27 | 707 | 16 | 5 | 2 | 1 | 2.26 | 0.71 | 0.28 | 0.14 | 3 [d] | 0 |
| 58 | STH 5.8506 | 22 | 644 | 30 | 13 | 4 | 3 | 4.66 | 2.02 | 0.62 | 0.47 | | |
| | | | | | | | STH 5.8506 a | | | | | 84 [bcd] | 0 |
| | | | | | | | STH 5.8506 b | | | | | 12 [d] | 1 |
| | | | | | | | STH 5.8506 c | | | | | 0 [d] | n/a |
| 59 | STH 5.8506/1 | 24 | 644 | 19 | 6 | 2 | 1 | 2.95 | 0.93 | 0.31 | 0.16 | 99 [bcd] | 0 |
| 60 | STH 5.8507 | 25 | 663 | 25 | 14 | 6 | 4 | 3.77 | 2.11 | 0.90 | 0.60 | | |
| | | | | | | | STH 5.8507 a | | | | | 11 [d] | 1 |
| | | | | | | | STH 5.8507 b | | | | | 0 [d] | n/a |
| | | | | | | | STH 5.8507 c | | | | | 0 [d] | n/a |
| | | | | | | | STH 5.8507 d | | | | | 0 [d] | 0 |
| 61 | STH 5.8508 | 23 | 593 | 33 | 9 | 2 | 1 | 5.56 | 1.52 | 0.34 | 0.17 | 63 [bcd] | 0 |
| 62 | STH 5.8509 | 23 | 644 | 13 | 4 | 2 | 1 | 2.02 | 0.62 | 0.31 | 0.16 | 0 [d] | 0 |
| 63 | STH 5.8512 | 28 | 1092 | 28 | 8 | 2 | 1 | 2.56 | 0.73 | 0.18 | 0.09 | 41 [cd] | 0 |
| 64 | STH 5.8513 | 27 | 727 | 13 | 3 | 1 | 0 | 1.79 | 0.41 | 0.14 | 0 | 0 [d] | 0 |
| 65 | STH 5.8514 | 30 | 726 | 17 | 4 | 0 | 0 | 2.34 | 0.55 | 0 | 0 | 0 [d] | 0 |
| 66 | STH 5.8518 | 29 | 728 | 24 | 3 | 0 | 0 | 3.30 | 0.41 | 0 | 0 | 0 [d] | 0 |

**Table 1** (*continued*)

| No | Genotype | No. of plants | No. of emasculated florets | No. of isolated embryos | No. of germinated embryos | No. of acclimated plants | No. of lines | Embryos/ florets (%) | Germinated embryos/ florets (%) | Acclimated plants/ florets (%) | Lines/ florets (%) | No. of seeds | No. of maize chromosome/ chromatin added to oat genome |
|---|---|---|---|---|---|---|---|---|---|---|---|---|---|
| 67 | STH 5.8518/1 | 23 | 582 | 30 | 10 | 6 | 4 | 5.15 | 1.72 | 1.03 | 0.69 | | |
| | | | | | | | STH 5.8518/1 a | | | | | 61 [cd] | 0 |
| | | | | | | | STH 5.8518/1 b | | | | | 51 [cd] | 0 |
| | | | | | | | STH 5.8518/1 c | | | | | 14 [d] | 0 |
| | | | | | | | STH 5.8518/1 d | | | | | 2 [d] | 0 |
| 68 | STH 5.8522 | 25 | 672 | 12 | 3 | 2 | 0 | 1.79 | 0.45 | 0.30 | 0 | 0 [d] | 0 |
| 69 | STH 5.8522/1 | 32 | 733 | 8 | 2 | 1 | 1 | 1.09 | 0.27 | 0.14 | 0.14 | 1 [d] | chromatin |
| 70 | STH 5.8523 | 31 | 742 | 9 | 1 | 1 | 0 | 1.21 | 0.13 | 0.13 | 0 | 0 [d] | 0 |
| 71 | STH 5.8525 | 23 | 692 | 39 | 12 | 5 | 0 | 5.64 | 1.73 | 0.72 | 0 | 0 [d] | 0 |
| 72 | STH 5.8526 | 33 | 682 | 16 | 5 | 0 | 0 | 2.35 | 0.73 | 0 | 0 | 0 [d] | 0 |
| 73 | STH 5.8528 | 23 | 575 | 17 | 2 | 0 | 0 | 2.96 | 0.35 | 0 | 0 | 0 [d] | 0 |
| 74 | STH 5.8529 | 29 | 818 | 15 | 0 | 0 | 0 | 1.83 | 0.00 | 0 | 0 | 0 [d] | 0 |
| 75 | STH 5.8530 | 20 | 532 | 19 | 9 | 2 | 2 | 3.57 | 1.69 | 0.38 | 0.38 | | |
| | | | | | | | STH 5.8530 a | | | | | 0 [d] | 0 |
| | | | | | | | STH 5.8530 b | | | | | 0 [d] | n/a |
| 76 | STH 5.8530/1 | 22 | 576 | 20 | 7 | 3 | 2 | 3.47 | 1.22 | 0.52 | 0.35 | | |
| | | | | | | | STH 5.8530/1 a | | | | | 74 [bcd] | 0 |
| | | | | | | | STH 5.8530/1 b | | | | | 56 [cd] | 0 |
| 77 | STH 5.8535 | 35 | 784 | 39 | 9 | 2 | 2 | 4.97 | 1.15 | 0.26 | 0.26 | | |
| | | | | | | | STH 5.8535 a | | | | | 19 [d] | 0 |
| | | | | | | | STH 5.8535 b | | | | | 1 [d] | chromatin |
| 78 | STH 5.8536 | 31 | 835 | 35 | 10 | 2 | 1 | 4.19 | 1.20 | 0.24 | 0.12 | 0 [d] | n/a |
| 79 | STH 5.8536/1 | 25 | 774 | 32 | 7 | 2 | 2 | 4.13 | 0.90 | 0.26 | 0.26 | | |
| | | | | | | | STH 5.8536/1 a | | | | | 1 [d] | chromatin |
| | | | | | | | STH 5.8536/1 b | | | | | 0 [d] | 0 |

Skrzypek et al. (2018), *PeerJ*, DOI 10.7717/peerj.5107

Peer J

**Table 1** (*continued*)

| No | Genotype | No. of plants | No. of emasculated florets | No. of isolated embryos | No. of germinated embryos | No. of acclimated plants | No. of lines | Embryos/ florets (%) | Germinated embryos/ florets (%) | Acclimated plants/ florets (%) | Lines/ florets (%) | No. of seeds | No. of maize chromosome/ chromatin added to oat genome |
|----|----------|---------------|----------------------------|-------------------------|---------------------------|--------------------------|--------------|----------------------|---------------------------------|--------------------------------|--------------------|--------------|---------------------------------------------------------|
| 80 | STH 5.8540 | 29 | 908 | 22 | 5 | 1 | 1 | 2.42 | 0.55 | 0.11 | 0.11 | 0 [d] | 0 |
| | Total | 2,314 | 57,515 | 2,129 | 601 | 210 | 138 | | | | | 4371 | |
| | Average | | | | | | | 3.58 | 0.99 | 0.34 | 0.21 | | |
| | Min. | | | | | | | 0 | 0 | 0 | 0 | | |
| | Max. | | | | | | | 9.69 | 2.63 | 1.83 | 1.46 | | |

**Notes.**

Significant differences between oat lines according to Duncan's test, $p \leq 0.05$, are marked with different letters.

*Ananiev, Phillips & Rines, 1998*) were used for the PCR reaction, which in successive cycles enabled the amplification of the 500 bp retrotransposon region of *Grande-1* and detection of the presence of maize chromatin in oat plants. For the PCR, the thermostable Taq polymerase (Fermentas, USA) was used.

After DNA isolation and dilution (50 µg/µl DNA), 5 µl of DNA was taken and introduced into reaction mixture using 0.5 µl of each two primers. The reaction mixture was placed in a thermocycler (2720 Thermal Cycler; Applied Biosystems, Foster City, CA, USA) and used the following thermal program: 1. Initial denaturation −94 °C, 5 min.; 2. Cyclic denaturation: 94 °C–30 s; 3. Starter connection—58 °C, 30 s; 4. Polymerization—72 °C—30 s; 5. Final Polymerization—72 °C, 5 min. Stages 2-4 were repeated cyclically 25 times. In the last stage the samples were cooled to 4 °C until removed from the amplification apparatus (*Ananiev, Phillips & Rines, 1998*).

The obtained products were mixed with 2.5 µL DNA Gel Loading Dye buffer (Thermo Fisher Scientific, Waltham, MA, USA) and then separated in 1.5% agarose gel with ethidium bromide (Sigma-Aldrich, St. Louis, MO, USA) in TBE buffer, under 90 V for 90 min. DNA markers of 100 bp to 1000 bp and concentration of 0.5 µg/µl (GeneRuler 100bp; Fermentas, Waltham, MA, USA) were used to estimate the length of PCR products. The image of electrophoretic separation was archived using the Imagemaster VDS gel reader (Amersham, Pharmacia Biotech, Piscataway, NJ, USA) and the Liscap Capture Application ver. 1.0. Electrophoretic gel analysis was performed using GelScan ver. 1.45, (Kucharczyk Electrophoretic Techniques, Warsaw, Poland). The plants identified as having a *Grande-1* retrotransposon fragment, which were also fertile and produced seeds, were used for genomic *in situ* hybridization (GISH).

## Cytogenetic analysis

The seeds of oat, maize, and oat-maize chromosome addition and DH lines were germinated for 2–3 days in Petri dishes on a wet filter paper. Seedlings with the root tips of 1.5–2 cm long were immersed in ice-cold water and incubated for 24 h at 4 °C. The procedure allowed to stop the cell division at metaphase through the spindle degradation, and to increase chromosome condensation. After 24 h, the root tips were fixed in a mixture of pure methanol and glacial acetic acid at a 3:1 ratio. The material was kept at 4 °C for 24 h, and stored in −20 °C until used.

Excised root tips (about 5 mm in length) were washed in citrate buffer (pH 4.8) for 15-20 min and then digested in a solution of 20% pectinase (Sigma, St Louis, MO, USA), 1% cellulose (Calbiochem, San Diego, CA, USA) and 1% cellulase 'Onozuka R-10′(Serva, Heidelberg, Germany) in a citrate buffer at 37 °C for 2 h. The meristems were dissected from the root tips and squashed in a drop of 45% acetic acid. Good quality preparations were frozen on dry ice. After freezing, the coverslips were removed and the slides were air-dried and stored at 4 °C until required.

In order to discriminate maize chromatin introgressions, the total genomic DNA of the maize cv. Waza was used as probe for GISH. Maize DNA was isolated from plants grown under greenhouse conditions (21 °C, photoperiod 8 h/16 h). Young leaves were frozen with liquid nitrogen and homogenized. DNA isolation was performed using the C-TAB method

(*Doyle & Doyle, 1987*). The concentration and purity of the isolated DNA was measured by NanoDrop 2000c (Thermo Scientific, Waltham, MA, USA) and the length of the DNA sequence fragments was checked on agarose gel.

Maize genomic DNA and 25S rDNA sequences that were used as additional chromosome markers were labeled by nick-translation using the digoxigenin-11-dUTP and rhodamine-5-dUTP (Roche), respectively. The GISH procedure was performed according to the protocol of *Hasterok et al. (2006)*. In order to verify the specificity of the gDNA probe and appropriate hybridization conditions, maize and oat root tips preparations were also used as positive and negative control in GISH, respectively. Hybridization mixture was prepared by adding appropriate volumes of 50% formamide, 10% dextran sulfate, 20 × SSC, water, SDS, and labeled DNA probes (∼500 ng/slide). Due to the high phylogenetic distance between oat and maize and related to that low degree of DNA sequence homology, the use of total genomic oat DNA to block non-specific hybridization was not necessary. Posthybridization stringent washes were carried out in 10% formamide in 0.1 × SSC at 42 °C, which is equivalent to 79% stringency. Immunodetection of digoxigenin-labeled probes was performed according to the standard protocols using FITC-conjugated anti-digoxigenin antibodies (Roche). The rhodamine probes were visualized directly. The preparations were mounted in VectaShield antifade (Vector Laboratories, Inc. Burlingame, CA, USA) containing 2.5 $\mu$m ml$^{-1}$ DAPI (Serva, Heidelberg, Germany). Hybridization signals were visualized and captured using fluorescence microscope Axio Imager Z2 (ZEISS, Germany) equipped with monochromatic camera AxioCam MRm (ZEISS, Oberkochen, Germany) and archived by ZEN blue program (ZEISS, Oberkochen, Germany). All images were processed uniformly and superimposed using Photoshop CS3 (Adobe).

## STATISTICS

Statistical analysis of embryo formation, DH and OMA lines as well as seeds production was performed using analysis of variance at $p \leq 0.05$ using the STATISTICA 12.0 software package (Stat-Soft, Inc., Tulsa, OK). For comparison of means for seeds number, Duncan's multiple test at the significance level of $p \leq 0.05$ was used. Correlation coefficient for seeds number dependently on maize chromosome/chromatin addition to oat genome was also calculated.

## RESULTS

### Oat DH lines and oat × maize hybrids formation

Analysis of variance indicated no significant differences between oat genotypes in the production of embryos per emasculated florets and lines per pollinated florets (Table 2). The efficiency of oat lines production by wide crossing with maize is shown in Table 1. Total of 57,515 florets from 80 oat genotypes were pollinated with maize that resulted in development of 2,129 embryos (Fig. 1A). Average embryo formation frequency per pollinated florets was 3.58%. Six hundred and one embryos germinated within three weeks of *in vitro* culturing (28.23% of all embryos) (Fig. 1B). Number of embryos produced by various genotypes varied from 0 to 106 (9.69% embryos per pollinated florets). Only one

**Table 2 Analysis of variance of haploid embryos formation and lines production depending on oat genotype as well as seeds production depending on oat hybrids.**

| Trait | SS | df | MS | MS for residual | F | p |
|---|---|---|---|---|---|---|
| Embryos formation | 217.13 | 79 | 2.895 | 34.464 | 0.084 | 0.751 ns |
| Lines production | 6.27 | 79 | 0.084 | 0.024 | 3.457 | 0.116 ns |
| DH/OMA seeds production | 49,347.9 | 1 | 49,347.9 | 2,607.8 | 18.923 | 0.000[*] |

**Notes.**

SS, sum of squares; df, degrees of freedom; MS, mean squares; ns, not significant $p \leq 0.05$.

*Significant at $p \leq 0.05$.

genotype out of eighty examined did not formed haploid embryos (STH 5.8428). Most embryos per pollinated floret were obtained from the genotypes: STH 4.4690 (9.69%), STH 4.4606 (7.22%), STH 4.4586 (6.68%), STH 5.8425 (6.54%) and STH 4.8448 (6.37%). Even if the genotype did not have the significant effect on the haploid plants production, 15 of 80 genotypes did not develop haploid plants. Total of 210 plants survived the acclimatization to the natural condition (Figs. 1C–1E). The procedure of chromosome doubling using colchicine reduced the number of obtained lines to 138. The highest number of lines (16) was obtained from the STH 4.4690 genotype. Plants from seven genotypes completely failed this process. The ploidy of the plants was measured by flow cytometry before and after application of colchicine (Fig. 2). Most plants doubled chromosomes number, except plants STH 4.4690 n, STH 4.4690 o, STH 5.8505 b, STH 4.8417 c and STH 5.8540.

## Identification of oat × maize hybrids

A total of 138 oat lines obtained by crossing with maize were tested for the presence of maize chromosomes. The presence of maize chromatin was indicated by amplification of the PCR product (500 bp) generated using primers specific for the maize retrotransposon *Grande-1* (GenBank accession number X97604; *Ananiev, Phillips & Rines, 1998*). Copies of the *Grande-1* are densely dispersed and located on each of the maize chromosomes. The 500 bp PCR product was identified in 66 oat plants. Figure 3 shows the exemplary agarose gel with 500 bp product corresponding to the maize retrotransposon *Grande-1*. Out of the 66 OMA lines containing maize chromatin, only 27 were fertile and produced seeds (Table 1). The 20 OMA lines that produced the highest number of seeds were then selected for cytogenetic analysis.

The seeds of two out of 20 OMA lines did not germinate. Among the remaining 20 lines, 8 lines (40%) had the character of additive lines, since GISH has detected one or more maize chromosomes in their cells. All of the analyzed plants possessed full complement of oat chromosomes. The number of detected maize chromosomes differed between the OMA lines (Table 1). They were usually smaller than the oat chromosomes and easy to discriminate (Fig. 4). In the plant STH 4.4690f, two pairs of chromosomes of varying size were found (Fig. 4A), whereas in STH 5.8504b only one maize chromosome was present (Fig. 4B). The majority of OMA lines possessed two maize chromosomes similar in size. In most of the lines, the detected chromosomes were labeled uniformly (Fig. 4C); however, in some, a banding pattern was visible that can potentially be used for the identification of maize addition (Fig. 4D). The banding pattern on both chromosomes in a given line was

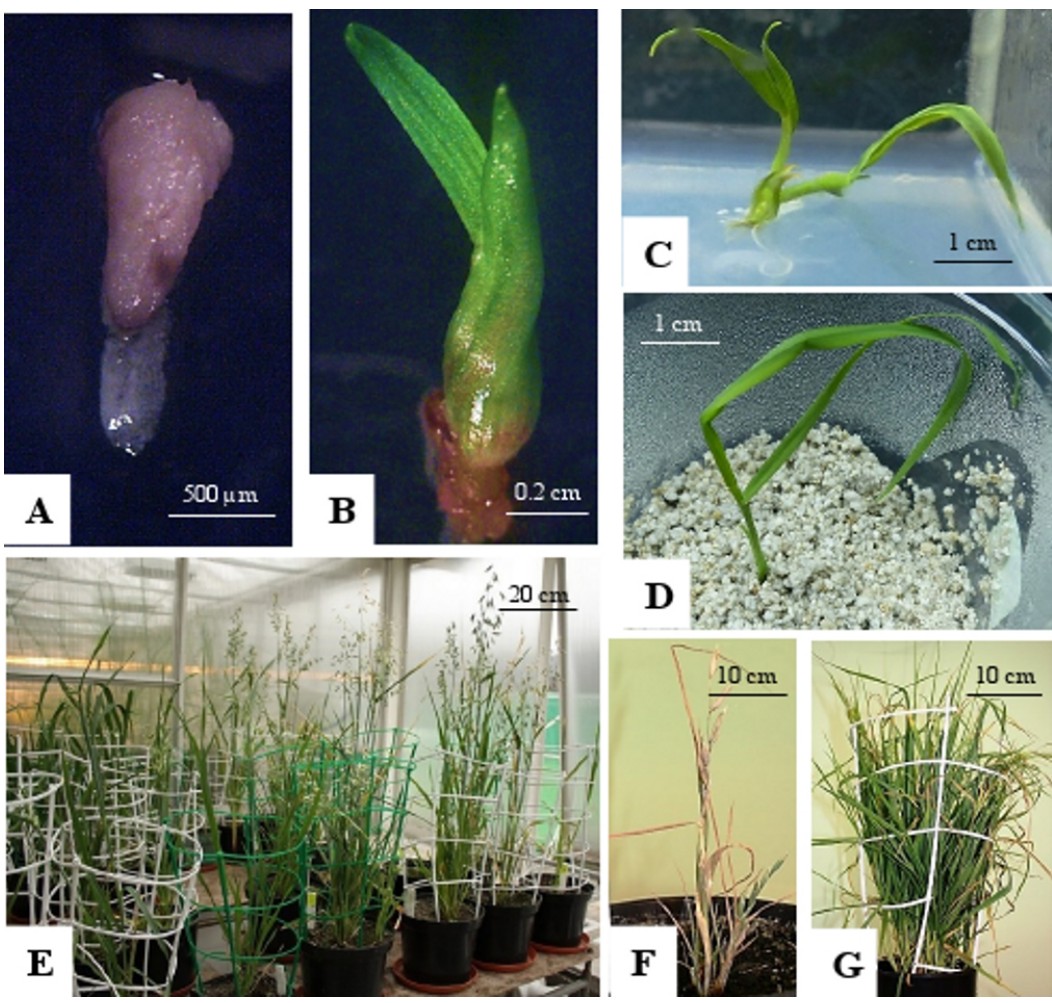

**Figure 1  Oat haploid embryo formed after crossing with maize.** (A); germinated haploid embryo on 190-2 medium (B); haploid plant on MS medium (C); acclimatization of haploid plant in perlite (D); DH plants in the greenhouse (E); panicles of OMA line STH5.8536/1, some of immature panicles are senesced (F); grassy OMA line STH5.8429 without panicles (G). Photo credit: Edyta Skrzypek.

similar, which might indicate that the added chromosomes are homologs. The presence of six 45S rDNA loci was detected in oat chromosomes (red hybridization signals, Fig. 4), but none of the added maize chromosomes in any of the lines carried 45S rDNA locus.

Twenty of the analyzed lines did not possess whole chromosomes of maize, but the introgression of maize chromatin was found in the oat chromosomes. A total of 12–14 hybridization signals for the maize gDNA probe were detected (Fig. 5). In all these lines, six of the maize gDNA hybridization signals colocalized with 25S rDNA sequence hybridization signals. In another pair of chromosomes, two strong signals were present in the pericentromeric positions on each of the chromosomes. Additionally, weaker maize gDNA signals were observed in one or two pairs of chromosomes of the lines STH 4.4690d (Fig. 5A), and STH 4.4606a (Fig. 5C), depending on the metaphase plate, and in one chromosome pair in the case of the line STH 4.4576 (Fig. 5B).

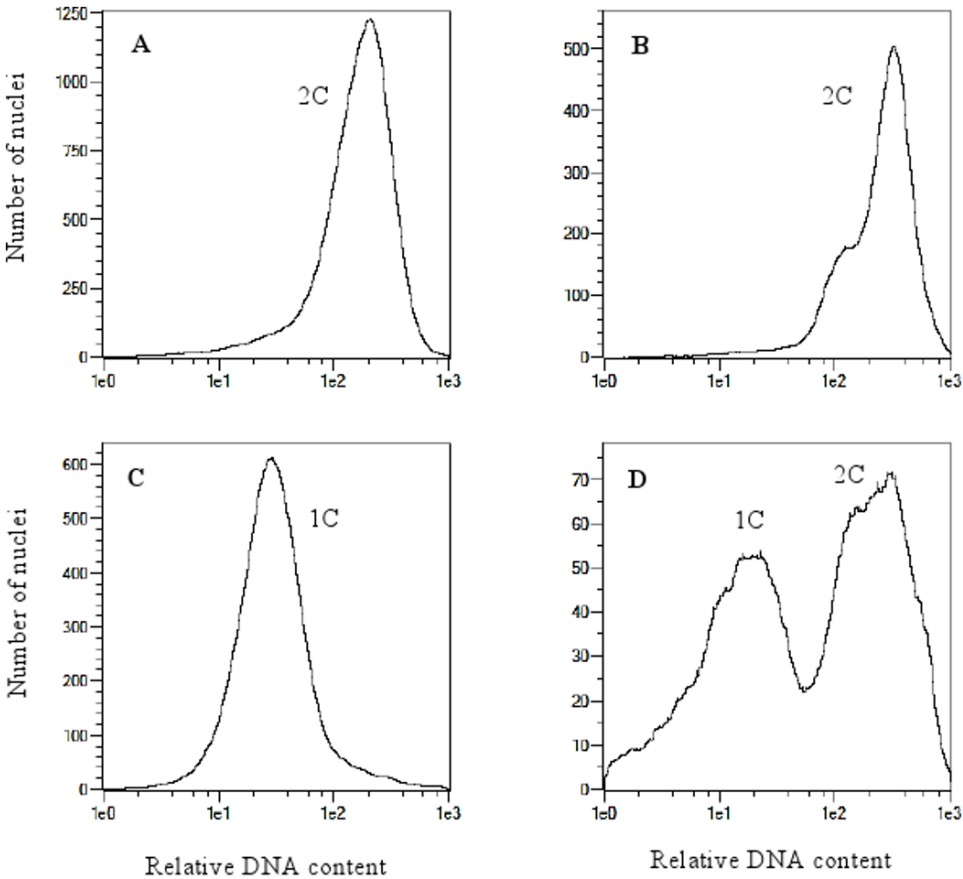

**Figure 2** Flow cytometry histograms of oat plants; (A) control 2n, (B) doubled haploid 2n, (C) haploid 1n and (D) mixoploid.

The colocalization of the gDNA and 25S rDNA signals may be related to the hybridization of the 25S rDNA sequence present in the maize genomic DNA to the same sequence in the oat genome. These sequences are highly conservative, hence the presence of these signals in the chromosomes of the analyzed lines. Additional hybridization signals found in 2–3 pairs of chromosomes, depending on the analyzed line, are presumably maize DNA sequences that have been incorporated into the oat chromosomes. Accurate determination of the origin of these sequences requires further analysis.

## Vigor of DH and OMA lines in their effectiveness in seeds production

The progeny of oat-maize hybrids showed some differences in morphology. Five of 66 hybrids were shorter in height, grassy type without panicles (Fig. 1G). Three of them had only a few short shoots with small panicles (Fig. 1F). Leaves' blades color and panicle shape were similar to their oat parent.

Analysis of variance showed significant variation in fertility in dependence on the addition of maize chromosomes to the oat genome (Table 1). OMA plants produced significantly less seeds than DH lines. Negative correlation coefficient (−0.279) for seeds

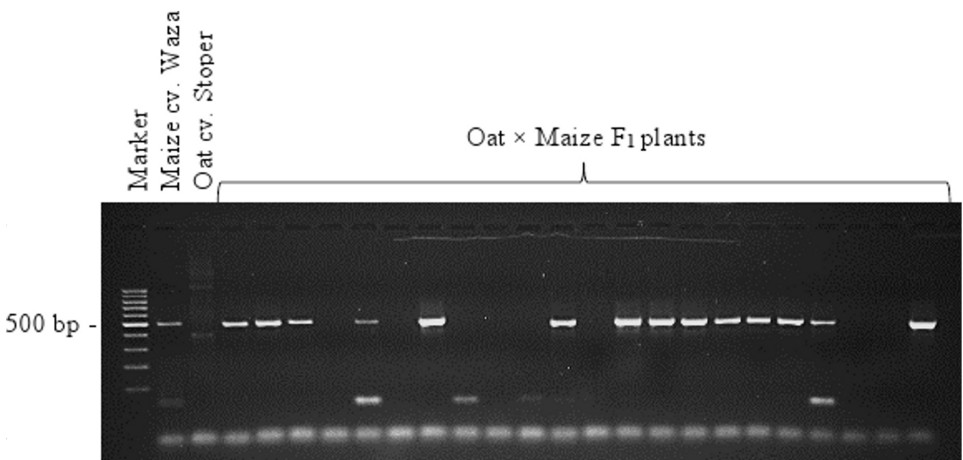

**Figure 3** **Identification of oat × maize F1 plants.** PCR products of genomic DNA of oat, maize, and a selection of 22 oat ×maize F1 plants shown after electrophoresis in 1.5% (w/v) agarose gel. Bands represent 500 bp DNA fragments that were amplified with marker *Grande-1*. Marker leader is shown in the first line. Maize cv. Waza specificity is shown by product presence in maize DNA (positive control) and absence in oat cv. Stoper DNA (negative control). The presence of retained maize chromosomes is indicated in 14 out of the 22 $F_1$ plant DNAs shown. Photo credit: Tomasz Warzecha.

production dependently on the maize chromosomes/chromatin added to the oat genome was observed.

Sixty-three fertile DH lines out of 72 which did not have an addition of maize chromosomes or chromatin produced seeds (Table 2). Nine DH lines belonging to various genotypes did not produce seeds. The number of seeds was from 1 to 343, dependently on the genotype. In total, 3,758 seeds were produced. The highest number of seeds (343 in total) was produced by the STH 5.8429 genotype.

Twenty-seven OMA lines (from 66 identified) were fertile and produced seeds ranging in number from 1–102 (Table 2). In total, 613 seeds were produced by OMA lines. As for DH lines, the number of seeds produced by OMA lines depended on genotype, however nearly 41% of OMA plants were sterile. The differences in the seed production were not dependent on the number of chromosomes added to the oat genome or introgression of chromosome fragments to the oat chromosomes. OMA lines with the addition of whole chromosomes produced seeds as well as the ones with the introgression of fragments of maize chromosomes. All DH and OMA lines were fertile and produced seeds in the subsequent generation.

## DISCUSSION

In wide hybridization in Poaceae, the complete or incomplete chromosomes elimination of one of the parents occurs (*Laurie & Bennett, 1986*; *Laurie & Bennett, 1988*; *Laurie, 1989*; *Rines & Dahleen, 1990*). There are several reports concerning the mechanisms of this process. In wheat × maize hybrids, Mochidaetal2004 found the partial addition of the spindle microtubules to the centromeres of maize. In wheat × maize crosses, haploid

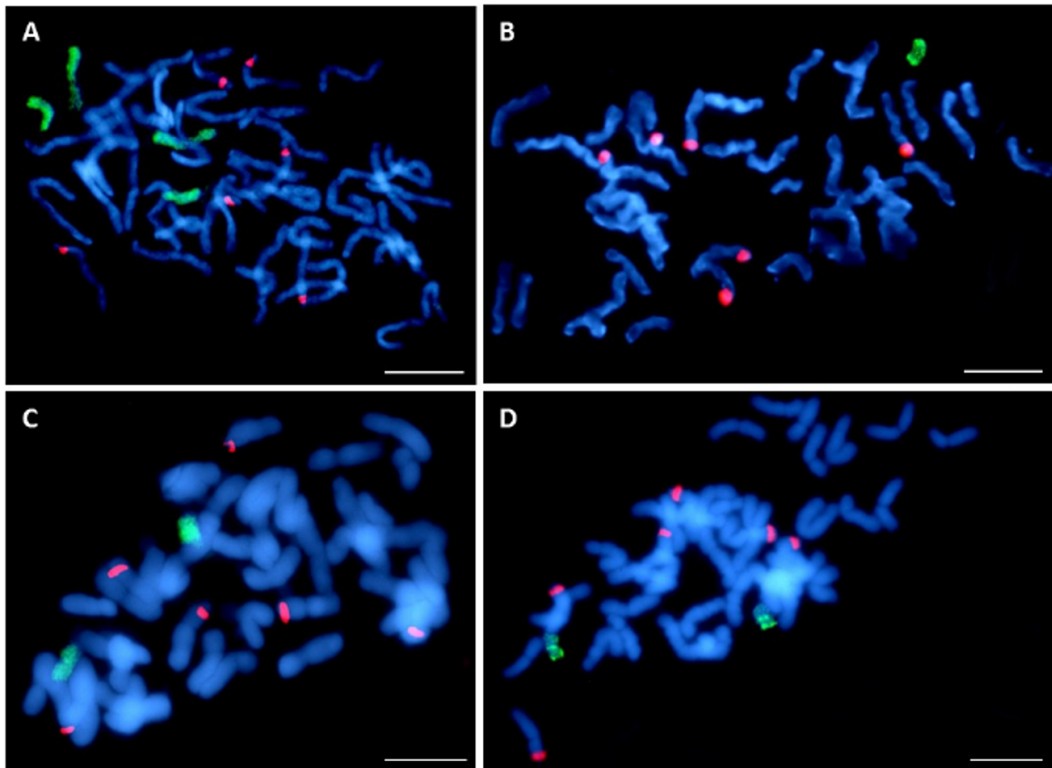

**Figure 4** **Visualization of added maize chromosomes in oat genome by genomic *in situ* hybridization (GISH).** (A) STH 4.4690f plant with tetrasomic addition of maize; (B) STH 5.8504b plant with monosomic addition of maize chromosome, (C) STH 5.8436b plant with disomic addition of maize chromosome. The maize chromosomes are labelled uniformly, (D) STH 4.4576 plant with disomic addition of maize chromosome. A banding pattern is visible on additional chromosomes. Maize gDNA is labeled with digoxigenin and detected anti-dig FITC (green fluorescence), rhodamine-5-dUTP—labeled 25S rDNA (red fluorescence) is used as an internal control of hybridization efficiency. Chromosomes are stained with DAPI (blue fluorescence). Scale bar: 10 μm. Photo credit: Dominika Idziak-Helmcke.

wheat plants are obtained as a effect of complete elimination of the maize chromosomes prior to embryo rescue. In a wheat × pearl millet hybrids cells, *Gernand et al. (2005)* detected delaying pearl millet chromosomes in anaphase and nuclear discarding of pearl millet chromatin in interphase. In the same wheat × pearl millet hybrids, *Ishii et al. (2010)* noticed chromosomes breaking and lack of consistency of the pearl millet chromosomes.

The wide-cross combinations between various species of Triticeae as the female and a wide variety of Panicoideae species including sorghum and pearl millet as the male, showed a similar pattern of hybrid zygote formation with uniparental loss of the donor pollen chromosomes during subsequent embryo development (*Rines et al., 2003*). Chromosomes elimination was reported in crosses of wheat (female) and *Hordeum bulbosum* L. with maize (*Zea mays*), pearl millet (*Pennisetum glaucum*), sorghum (*Sorghum bicolor*), *Coix lacryma-jobi*, and *Imperata cylindrica* (*Barclay, 1975*; *Laurie & Bennett, 1986*; *Laurie & Bennett, 1988*; *Laurie & Bennett, 1989*; *Laurie, 1989*; *Inagaki & Mujeeb-Kazi, 1995*; *Mochida & Tsujimoto, 2001*; *Komeda et al., 2007*; *Ishii et al., 2010*). The cells of *H. bulbosum* fertilize

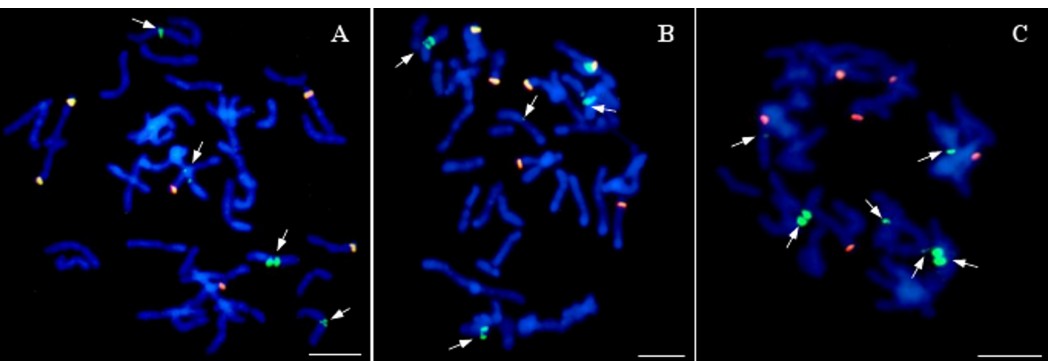

**Figure 5** **Visualization of added fragments of maize chromosomes in oat genome by genomic *in situ* hybridization (GISH).** (A) STH 4.4690d; (B) STH 4.4576; (C) STH 4.4606 F1 plants. The arrows point to maize introgressions into oat chromosomes (green fluorescence). Yellow signals result from colocalization of hybridization signals for maize gDNA (green fluorescence) and 25S rDNA (red fluorescence). Chromosomes are stained with DAPI (blue fluorescence). Scale bar: 10 μm. Photo credit: Dominika Idziak-Helmcke.

the cells of wheat typically, but in hybrid embryogenesis the chromosomes of *H. bulbosum* are progressively extruded from the hybrid nucleus in the anaphase–telophase transition. The process is proceeded by micronuclei and condensed chromosomes formation (*Zenkteler & Straub, 1979*).

Genomic *in situ* hybridization (GISH) is a modification of fluorescence *in situ* hybridization (FISH) in which the total genomic DNA of a given species is used as a probe to hybridize with the complementary DNA in a cytogenetic preparation (*Schwarzacher, 2003*). This technique allows for the genome-specific chromosome labeling in the cells of allopolyploid or hybrid species. It has been widely used in plant studies to identify putative parental genomes in the interspecific or intergeneric hybrids or allopolyploids (for the review see *Chester et al., 2010*; *Silva & Souza, 2013*), to study the evolution of polyploid genomes (*Song et al., 1995*; *Gaeta et al., 2007*; *Majka et al., 2018*; *Tan et al., 2017*), or to detect the introgression of alien chromatin (*Schneider et al., 2005*; *Tan et al., 2005*). It has also become a valuable tool to analyze the meiotic recombination and behavior of chromosomes in the hybrids (*Zwierzykowski et al., 2008*) as well as translocation (*Kruppa et al., 2013*), addition (*Molnar-Lang et al., 2000*; *Ji & Chetelat, 2007*), or substitution (*Pan et al., 2017*) lines.

In the GISH-based studies, an excess of unlabeled blocking genomic DNA derived from the other parental species is often necessary in order to avoid non-specific cross-hybridization. Alternatively, the genomic DNAs of all parental species can be labeled differentially and hybridized simultaneously to the chromosomes of a hybrid/allopolyploid. In the studies of the genus *Paphiopedilum*, various ratios of blocking DNA has been applied in order to distinguish between the component genomes of the analyzed hybrid species, depending on the phylogenetic distance between the putative parents (*Lee, Chang & Chung, 2011*). In the case of a very close phylogenetic relationship between the parental species, even very high ratios of the blocking DNA are often not sufficient to distinguish

the chromosomes of each donor genome (*Xiong et al., 2006*; *Lee, Chang & Chung, 2011*). On the other hand, in the intergeneric hybrids and allopolyploids, GISH can be applied successfully even without blocking DNA (*Tang et al., 2011*). In our studies, the large genetic distance between maize and oat allowed us to specifically label and distinguish maize chromatin with no blocking DNA from oat. It should be noted, though, that the ease of discriminating the genomes of genetically distant species is usually coupled with difficulty in generating the crosses between them.

Although the GISH results allow us to unambiguously detect maize chromatin, the information provided by this method is not sufficient to identify which particular maize chromosome was added to the oat genome. The observation that some chromosomes in were labeled uniformly, whereas other displayed clearly a banding hybridization pattern, indicates that different maize chromosomes were retained in various OMA lines. The banding pattern probably reflects varying distribution of highly repetitive sequences in the chromosomes of maize since these sequences usually constitute the most of genomic DNA used as a probe. In the studies by *Rines et al. (2009)*, the identification of the maize chromosomes in the produced OMA lines was achieved using maize chromosome-specific simple sequence repeat (SSR) markers. An alternative, cytogenetic approach would require performing a FISH experiment with several different types of repetitive sequences as probes, labeled with different fluorophores and hybridized simultaneously in order to create unique banding pattern on added maize chromosomes. Comparing the results with the FISH on maize as a control would allow to discover the identity of the chromosome addition. Such multicolor-FISH method utilizing tandemly repeated DNA sequences has been applied before to maize chromosomes and permitted to distinguish each of the 10 chromosomes (*Kato, Lamb & Birchler, 2004*).

Oat (*Avena sativa* L.) haploid plants can be induced by hybridization either with maize (*Rines et al., 1996*; *Sidhu et al., 2006*; *Marcińska et al., 2013*; *Nowakowska et al., 2015*; *Warchołet al., 2016*) or with pearl millet (*Rines et al., 1997*; *Ishii et al., 2013*). In oat and maize crosses, sometimes it happens that some maize chromosomes are not extruded during embryogenesis, and they ultimately stabilized and acted as oat chromosomes in mitosis (*Okagaki et al., 2001*). About one-third of the plants obtained from oat × maize crosses contain one or more maize chromosomes added to a complete set of oat chromosomes (*Rines et al., 2003*; *Rines et al., 2009*). Over the past few years, all ten maize chromosomes have been recovered as single chromosome additions to a haploid oat complement.

*Riera-Lizarazu, Rines & Phillips (1996)* after oat pollination by maize obtained 30 plants (one third of all plants) with retained one to four maize chromosomes, as a result of partial maize chromosomes elimination. Hybrid plants, in which some cells contained maize chromosomes and other did not, also were found. Authors explained that maize chromosomes elimination in crosses between oat and maize is more gradual than e.g., in wheat and maize. In the study by *Kynast et al. (2012)*, hybrids of oat and maize were euhaploids with a complete set of oat chromosomes without maize chromosomes, and aneuhaploids with a complete set of oat chromosomes and various number of retained maize chromosomes. Some of these haploids formed seeds to fifty percent of their spikelets. However, little is known about hybrids fertility *Kynast et al. (2012)* noticed that the

total seeds of $F_1$ hybrids generation with maintained maize chromosomes seemed to be meaningfully influenced by the number of retained maize chromosomes as well as by interaction of maize chromosomes with oat chromosomes. Also, the environmental conditions during $F_1$ hybrids cultivation appeared to had an impact on fertility. Based on chromosome observations and marker analyses in $F_1$ generation of oat × pearl millet crosses, *Ishii et al. (2015)* showed that the seedlings were full hybrids with all oat and pearl millet chromosomes. Although the hybrids of oat with pearl millet developed, they became necrotic in light condition.

In our experiment we found 53% of haploid plants and 47% of hybrids obtained after oat × maize crosses. The hybrids retained one to four maize chromosomes or fragments of maize chromosomes added to the oat chromosomes. Nearly 41% of hybrids was infertile. In the study by *Kynast & Riera-Lizarazu (2011)*, offspring of oat × maize consisted of 50–66% haploid oat plants and 34–50% partial hybrids. Haploid oat and partial hybrids with one to three maize chromosomes were incompletely fertile. Partial hybrids with more than three maize chromosomes had were smaller than haploid plants and sterile.

In our experiment seeds production depended on oat genotype and retained maize chromosomes in oat genome. Among 72 oat DH lines, nine did not produce seeds, whereas in the group of 66 hybrids—39 were infertile. Seeds produced even hybrid with retained four maize chromosomes. The hybrids also showed differences in morphology. Some of them were visibly shorter than maternal and DH plants. In contrast to *Kynast et al. (2001)*, they did not show differences in the color of leaf blade and panicle shape.

### Funding
The research was funded by the National Centre for Research and Development project no. PBS3/B8/17/2015. The funders had no role in study design, data collection and analysis, decision to publish, or preparation of the manuscript.

### Grant Disclosures
The following grant information was disclosed by the authors:
The National Centre for Research and Development: PBS3/B8/17/2015.

### Competing Interests
Zygmunt Nita and Krystyna Werwińska are employed by Plant Breeding Strzelce Ltd., PBAI Group.

### Author Contributions
- Edyta Skrzypek and Tomasz Warzecha conceived and designed the experiments, performed the experiments, analyzed the data, contributed reagents/materials/analysis tools, prepared figures and/or tables, authored or reviewed drafts of the paper, approved the final draft.
- Angelika Noga performed the experiments, analyzed the data, approved the final draft.

- Marzena Warchoł performed the experiments, authored or reviewed drafts of the paper, approved the final draft.
- Ilona Czyczyło-Mysza, Kinga Dziurka, Izabela Marcińska, Kamila Kapłoniak, Agnieszka Sutkowska, Krystyna Werwińska, Magdalena Rojek and Marta Hosiawa-Barańska performed the experiments, approved the final draft.
- Zygmunt Nita performed the experiments, contributed reagents/materials/analysis tools, approved the final draft.
- Dominika Idziak-Helmcke performed the experiments, analyzed the data, authored or reviewed drafts of the paper, approved the final draft.

## Data Availability

The raw data are provided in a Supplemental File.

## Supplemental Information

Supplemental information for this article can be found online at http://dx.doi.org/10.7717/peerj.5107#supplemental-information.

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
