# Peer review of "Complex characterization of oat (Avena sativa L.) lines obtained by wide crossing with maize (Zea mays L.)"

_PeerJ, doi:10.7717/peerj.5107_

## Round 0.1 · original submission · Minor Revisions

Please address the reviewer comments; I especially agree that the long list (lines 116-128) could be omitted - in the methods state that the lines used are listed in Table 2 (not in another supplemental table), and I agree that you should combine tables 2, 3 and 4.

·

Basic reporting

no comment

Experimental design

Correct.

Validity of the findings

no comment

Additional comments

Please find below the review of the manuscript: “Complex characterization of oat (Avena sativa L.) lines obtained by wide crossing with maize (Zea mays L.)”. The topic of the manuscript is interesting from practical and theoretical point of view.
High level of homozygosity possessed by DH lines is a feature highly desirable for durability of certain characteristics of new cultivars. The advantage of this technique is the DH lines could be generated in one round of reproduction. Because of this reason the technique allows to shortened time of producing new cultivars. Among couple of method to generate DH lines is wide crossing. For oat this method is most effective to get oat (Avena sativa) haploids (actually polyhaploids since oat is hexaploid) and then DH lines with application of maize pollen. Therefore the manuscript is interesting for plant breeders because is focused on efficiency of getting DH lines. Typically in wide crossing chromosomes from pollen donor are completely eliminated in the early stage of embryogenesis, but in oat one or more maize chromosomes could be retained. Interestingly oat can form stable and fertile hybrids with maize (OMA lines). In the study presented in this manuscript Authors also look at the lines which retained fragments or whole maize chromosomes because it may result in morphological changes and affect the yield of oat, and more generally have impact on fertility.
Although I highly appreciated the topic and the research effort made by Authors I have several minor comments presented bellow.

Detailed comments:

Introduction
Extend literature review to some information concerning mutants of CENH3 centromers, as this method becomes a new tool for haploids production.

Material and methods
Explain the abbreviation NAA and KIN.
Was only roots of haploid plants treated with colchicine for chromosome doubling?
Statistics section is poor. Please, rewrite this.

Results
Information about correlation between observed traits would be interested.

References
Check carefully cited literature e.g. Rines et al. 2009 is listed twice, and some literature is incomplete e.g. Majka et al. 2017, Chaudhary et al. 2013.

Tables
Table 1 needs sum of squares and mean squares for residual.
Table 2 needs LSD or HSD values for observed traits.

Figures.
The Authors use large number of DH lines and gave the encoded name of each one, They also mention cv. Waza used as pollinator, but I found in Figure 1. that Authors used in PCR amplification with Grande 1 primers, maize in line 2 and oat in line 3. Could you please specify in the figure the exact genotype e.g. for maize cv. Waza and in the description of Figure 1, cv. Waza – positive control , and for oat (actually we don’t know what was that, therefore the reader could be lost or not sure what genotype is there so, I suggest in description of line 3 to write cv. Stoper (I assumed that it was this genotype, but it must be clear that there is no oat DH line), and in the description of Figure 1 cv. Stoper – negative control. I also suggest to change the legend above the electrophoresis: “Oat x Maize F1 plants” to “Oat x maize F1DH plants”

In my opinion the manuscript covers an interesting and valuable scientific information about production and characterization of oat doubled haploids and oat x maize hybrids. Therefore I think it meets aims and scopes of the PeerJ journal, and I recommend the manuscript for publication but after minor revision.

Reviewer 2 ·

Basic reporting

See General comments

Experimental design

See General comments

Validity of the findings

See General comments

Additional comments

The OMA lines have acted as important materials using not only in the new resource development but also in the fundamental studies such as centromere biology and epigenetic dynamics of imported phenotypes in hybrids. In this study, the authors developed a series of new OMA lines and conducted basic cytogenetic study to characterize the karyotype for those lines. However, disappointed, the story is fairly simple and straightforward. The manuscript was not composed in a professional way and should be rewritten completely.
1\Actually, there are numerus literatures published based on studes using OMA lines. However, only few “old” papers were mentioned by authors. The authors should rewrite the section in Introduction to make an extensive introduction on the utility of the OMA lines.
2\Lines 116, the genotypes should be show as a supplemental table other than as a long-list in the manuscript.
3\ In Methods, the method descriptions should be shortened or deleted if method was conducted as published method and seems no any modifications were made.
4\ Table 2,3,4 looks so redundant and unreadable. They should be rearranged to make them concise.
5\line 271-279 those should be moved to the Method.
6\lines 281 “whole maze chromosome” What is that mean?
7\lines 293 Figures should be shown here to support the description of 45S rDNA.
8\ lines 294 to 301 states that “A total of 12-14 hybridization signals for the maize gDNA probe were detected (Figure 4d)”, which doesn't match what the figure actually shows. Please rectify all the confusing statement in the whole paragraph.
9\lines 315-316 This paragraph contains only one sentence, it is so strange.
10\ Scale bars for every separate image should be added in Figure 1.
11\ lines 311, the noun progeny can be countable or uncountable. In more general, commonly used, contexts, the plural form will also be progeny.

---

## Round 0.2 · accepted · Accept

Thank you for addressing all the rveviewer comments; your paper is now accepted.

#